# The Evaluation of Cartilage Regeneration Efficacy of Three-Dimensionally Biofabricated Human-Derived Biomaterials on Knee Osteoarthritis: A Single-Arm, Open Label Study in Egypt

**DOI:** 10.3390/jpm13050748

**Published:** 2023-04-27

**Authors:** Mohamed M. Abdelhamid, Gaber Eid, Moustafa H. M. Othman, Hamdy Ibrahim, Dalia Elsers, Mohamed Elyounsy, Soon Yong Kwon, Minju Kim, Doheui Kim, Jin-Wook Kim, Jina Ryu, Mohamed Abd El-Radi, Tarek N. Fetih

**Affiliations:** 1Department of Orthopedic Surgery, Faculty of Medicine, Assiut University, Assiut 71511, Egypt; 2Department of Orthopedic Surgery, Faculty of Medicine, Azhar University, Assiut 71511, Egypt; 3Department of Radiology, Faculty of Medicine, Assiut University, Assiut 71511, Egypt; 4Department of Pathology, Faculty of Medicine, Assiut University, Assiut 71511, Egypt; 5Department of Plastic Surgery, Faculty of Medicine, Assiut University, Assiut 71511, Egypt; 6Department of Orthopedic Surgery, College of Medicine, Catholic University of Korea, Seoul 06591, Republic of Korea; 7Department of Research and Development, ROKIT Healthcare, Seoul 08514, Republic of Korea

**Keywords:** articular cartilage, 3D bioprinting, adipose tissue, allogenic hyaline cartilage, knee, osteoarthritis

## Abstract

Full thickness cartilage defects in cases of knee osteoarthritis are challenging in nature and are difficult to treat. The implantation of three-dimensional (3D) biofabricated grafts into the defect site can be a promising biological one-stage solution for such lesions that can avoid different disadvantages of the alternative surgical treatment options. In this study, the short-term clinical outcome of a novel surgical technique that uses a 3D bioprinted micronized adipose tissue (MAT) graft for knee cartilage defects is assessed and the degree of incorporation of such graft types is evaluated via arthroscopic and radiological analyses. Ten patients received 3D bioprinted grafts consisting of MAT with an allogenic hyaline cartilage matrix on a mold of polycaprolactone, with or without adjunct high tibial osteotomy, and they were monitored until 12 months postoperatively. Clinical outcomes were examined with patient-reported scoring instruments that consisted of the Western Ontario and McMaster Universities Arthritis Index (WOMAC) score and the Knee Injury and Osteoarthritis Outcome Score (KOOS). The graft incorporation was assessed using the Magnetic Resonance Observation of Cartilage Repair Tissue (MOCART) score. At 12 months follow-up, cartilage tissue biopsy samples were taken from patients and underwent histopathological examination. In the results, at final follow-up, the WOMAC and KOOS scores were 22.39 ± 7.7 and 79.16 ± 5.49, respectively. All scores were significantly increased at final follow-up (*p* < 0.0001). MOCART scores were also improved to a mean of 82.85 ± 11.49, 12 months after operation, and we observed a complete incorporation of the grafts with the surrounding cartilage. Together, this study suggests a novel regeneration technique for the treatment of knee osteoarthritis patients, with less rejection response and better efficacy.

## 1. Introduction

Knee osteoarthritis (KOA) is the most common progressive disease-causing musculoskeletal disability for patients in the world [1]. Its pathogenesis includes detrimental alterations of the joint, including synovial inflammation, cartilage matrix degradation, and following exposure of subchondral bone tissues. The articular cartilage is a complex connective tissue that provides a smooth, lubricated, friction-reducing surface [2]. Articular cartilage can withstand repeated tremendous forces but does not have the ability to heal even after minor injuries because of its avascular, aneural, and alymphatic characteristics [3,4]. It consists of dense extracellular matrices, with chondrocytes derived from mesenchymal stem cells during the development process. The extracellular matrices contain predominantly water, collagen fibers, and proteoglycans, along with non-collagenous proteins. Of all, type II collagen (COL2) is the most abundant form of collagen fibers in the chondral extracellular matrix [2,5].

Because of its inability to repair itself, surgical intervention is often required to resolve the progression of cartilage degeneration in affected patients [4]. However, surgical treatment methods remain challenging because of their inability to regenerate hyaline-like cartilages.

Current surgical treatments for knee osteoarthritis cases that are non-responsive to medical treatment include arthroscopic lavage, cartilage management techniques, unloading of the diseased part of the joint such as high tibial osteotomy, or replacing the joint surface with artificial inserts such as unicondylar or total knee replacement [6].

Although the last therapeutic option has been shown to relieve pain and improve mobility in people, it may carry several complications such as stiffness, instability, aseptic loosening, infection, prosthesis failure, and malalignment [7].

Several techniques have been described for cartilage lesions management, such as debridement, microfracture (MFx), bone marrow stimulation, cell-based therapies, and auto or allogenic cartilage implantation [8,9]. Techniques such as microfracture aim to recruit cells (e.g., mesenchymal stem cells (MSCs) from the underlying bone marrow, while cell-based therapies such as matrix-induced autologous chondrocyte implantation (MACI) focus on scaffolds to elicit tissue generation from donor cells [10].

In the last few decades, great ambitions were laid upon regenerative medicine as an alternative method for traditional surgical procedures. The idea is that the regenerated cartilage tissue should behave chemically and mechanically as the native articular cartilage [11]. The most widely used method for articular cartilage regeneration is autologous chondrocyte implantation (ACI). The procedure was first introduced by a Swedish group in 1994. The procedure consisted of the removal of small biopsies of cartilage (200/300 mg) from less loaded areas of the knee. Isolated chondrocytes were cultured in monolayer, and after 2–3 weeks, harvested as a suspension for implantation. The suspension was then injected into the defect under a periosteal flap derived from the proximal medial tibia (first-generation ACI) [12]. Last-generation ACI, also referred to as MACI, comprises the seeding of in vitro expanded chondrocytes onto a three-dimensional (3D) scaffold. The engineered tissue is subsequently trimmed to the size of the defect and implanted with fibrin glue fixation [13]. To our knowledge, it was used in humans for knee OA only once in 2010 by Means et. al [14]. However multiple limitations have been reported, such as insufficient intercellular binding to existing cell/tissues and the deterioration of healthy cartilage [15], and the ACI was considered contraindicated in OA, since the degenerative microenvironment may cause the implanted chondrocytes to undergo undesired dedifferentiation or apoptosis, therefore undermining the efficacy. In addition, the ACI procedure is considered to be long, complicated, and expensive. It also needs two hospitalization events for the patient—one for the biopsy harvest, and one for the implantation.

In the last few decades, attempts on the regeneration of articular cartilage have been performed through the use of mesenchymal stem cells (MSCs) injected intra-articularly. They are easy to isolate and do not imply ethical problems [16]. Lately, stem cells derived from subcutaneous adipose tissues (ASCs) have been considered as a less invasive procedure and are suitable for application in regenerative therapies. The advantage over marrow SCs is that ASCs are present at a higher percentage (500-fold more) [17]. The combination of stem cells and the chondron (hyaline cartilage matrix) was used in a goat model in 2013 [18].

Despite the clinical outcomes obtained with each of these procedures, only osteochondral autograft and allograft implantation partially succeeded to regenerate the intact architecture and the similar composition of the native, mature, COL2-rich hyaline cartilage [19,20]. The graft implantation is the considered as the safest and most cost-effective therapy because of less tissue rejection responses. However, the osteochondral grafts have limitations, particularly regarding their poor supplies, donor site morbidity, and contamination risks.

Three-dimensional (3D) printing, and specifically 3D bioprinting has been used for the fabrication of blood vessels, heart, liver, neural tissue, and cartilage [21,22]. The self-assembly and self-organizing capabilities of cells have been delivered through the applications of distinct bioprinting techniques [23,24,25,26]. Bioprinting may provide a means for stem cell implantation into host tissues in a very organized and complex manner [27]. Direct cartilage repair with engineered tissue closely mimicking native cartilage to the site of the lesion without any additional damage to the existing healthy tissue is therefore attractive. The ideal implantation is expected to integrate with existing native cartilage and to repair lesions of various sizes and thicknesses.

Since adipose tissue contains a large number of mesenchymal stem cells (MSCs), it is one of the most preferred tissues for collecting autologous MSCs. Unlike bone marrow MSCs, adipose MSCs can be isolated in large quantities with minimal morbidity and discomfort [28,29]. The frequency of MSCs in adipose tissue is in the order of 1 in 100 cells, about 500-fold more than that found in bone marrow [30]. In addition, as mechanically micronized adipose tissues (microfat) show abundant MSCs and pericytes preserving the stromal vascular fraction (SVF) and they are relatively easier to process in the operation room compared to enzymatic methods, it has been suggested that microfat is used as regeneration source for wound healing [31].

Therefore, our aim is to evaluate the clinical and radiological outcomes of using a solidified autograft prepared from MAT with allogenic hyaline cartilage powder implants that are prepared on a 3D printed polycaprolactone (PCL) scaffold, for the treatment of localized articular cartilage defects of KOA.

## 2. Materials and Methods

### 2.1. Patient Enrollment Criteria

Ten patients were finally enrolled in the study. The patients were aged between 18 and 60 years of age and attended the outpatient clinics of Assiut University Hospitals complaining of knee joint pain and swelling. All patients did not have any previous knee surgery. Patients with a body mass index of less than 35 were included. All patients had no major ligamentous instability on clinical examination. The patients were assessed clinically using the Knee Injury and Osteoarthritis Outcome Score (KOOS), and The Western Ontario and McMaster Universities Arthritis Index (WOMAC) scores.

Only patients with normal laboratory investigations for coagulation, and for renal and hepatic functions were included in the study. The study was approved by the ethics committee of Assiut University; all patients signed an informed consent for participation, and females with childbearing potential agreed to practice adequate birth control to prevent pregnancy during the study duration. All patients with inflammatory arthritis such as rheumatoid or gouty arthritis were excluded from the study. Patients with factors that could compromise the printed implant incorporation; such as patients who received immunosuppressive drugs, chemotherapy, or radiotherapy; and patients who had systemic or intra-articular steroid injections, or any form of intra-articular therapy in the last 2 months before the start of the study; as well as patients who smoke excessively or abuse alcohol were also excluded. Any patient in whom liposuction could cause any problem was not included in the study.

Imaging studies included plain antero-posterior erect, and lateral views of the knee joint to evaluate the cartilage defect, in addition to a three joints standing antero-posterior view of the lower limb to assess frontal plane alignment. Patients included in the study had normal lower limb alignment or a varus deformity of less than 10 degrees (mechanical axis). Magnetic resonance imaging (MRI) of the knee was used to assess the site, size, and thickness of the articular cartilage defects. All the patients in the study had a single focal, full-thickness cartilage defect as a result of aging, trauma, or degenerative diseases, which measured 2 to 10 cm^2^.

Patients included in the study were divided into two groups based on the frontal alignment of the involved knee. Patients in Group A had normal alignment or varus deformity of less than 5 degrees and underwent cartilage management techniques only, while patients in Group B had varus deformity of more than 5 degrees (5–10). This group had high tibial osteotomy in addition to the cartilage management in order to correct alignment and to enhance printed implant incorporation. Finally, a total of 10 participants were selected (5 male and 5 female), aged between 18 and 60 years old. The information of the participants enrolled in this study is summarized in Table 1.

### 2.2. Surgical Procedures and Biomaterial Ink Preparation

All procedures were performed in supine positions and under spinal anesthesia. A thigh tourniquet was used. All patients underwent knee arthroscopy using standard antero-lateral and anteromedial portals. The complete evaluation of all intra-articular structures was performed, followed by an accurate localization of the cartilage defect, and a measurement of its dimensions using a graded probe. Sometimes the switching of the viewing and working portals was performed in order to complete accurate measurements of all the defect margins. The cartilage lesion was graded according to the International Cartilage Repair Society (ICRS) system. The global status of the articular cartilage of the knee was evaluated carefully, and any minor meniscal pathology was dealt with in the form of shaving or trimming of the unstable or frayed edges.

After confirmation of the full thickness defect and the precise recording of the dimensions, liposuction was performed through a minor 1 cm incision on the abdomen to obtain an adequate amount of emulsified adipose tissues (50–60 cc).

The obtained fat was processed on the side table via micronization into liquid form. The MAT from autologous adipose tissue was mixed with allogenic hyaline cartilage powder (100 mg in 1 cc of MAT). The prepared bioink was delivered under sterile conditions into the 3D bioprinter (Dr. INVIVO printer, ROKIT Healthcare, Inc, Seoul, Republic of Korea). The first station prints a scaffold of the PCL according to the size, depth, and shape of the ulcer that was fed to the device using a specific software. MAT was printed on the 3D printed scaffold of the PCL from the second station of the Dr. INVIVO 3D printer. The printed graft was mixed with fibrin glue and left to solidify a little via gelatinization to transform into an implant with the exact same measured dimensions of the ulcer. Right before the implantation process, the solidified patch was detached from the PCL scaffold.

During the preparation of the implant, the surgeon performs limited arthrotomy over the defect site for Group A patients to complete the preparation of the defect site into a stable rim of cartilage, or to perform a medial wedge opening high tibial osteotomy using locked plate via an anteromedial approach for Group B patients; the approach is extended to a limited arthrotomy over the defect site. After obtaining the fashioned implant from the printer, it is fixed into the defect site using fibrin glue, and the stability of the implant fitted in the site is evaluated using a gentle range of motion cycling of the joint. The wound is closed in layers, and dressing is applied. The procedure is visualized in Figure 1.

### 2.3. Postoperative Rehabilitation Programs and Follow-up Procedures

Patients are encouraged to perform passive range of motion (ROM) exercises from the first day postoperative, and quadriceps isometric exercises from the second day of surgery, and active exercises are delayed to the second week of surgery. The patient performs a non-weight bearing program for 6 weeks, then partial weight bearing for 4 weeks. Follow-up for instructions and rehabilitation programs are performed at 2 and 6 weeks after surgery. All patients are evaluated clinically using the three scoring systems, as well as through plain and MRI studies at regular visits 3, 6, and 12 months postoperatively (3 M, 6 M, and 12 M, respectively). The incorporation of the implant into the defect site on MRI images is evaluated using The MOCART score (Magnetic Resonance Observation of Cartilage Repair Tissue) [32].

After about 12 M, postoperative second look arthroscopy and biopsy from the implant-native cartilage interface were performed using a two mm. trephine tool, for selected patients during the metal removal of the osteotomy plate or if the patient had a complaint about his/her knee. The biopsy was sent for histopathologic examination.

### 2.4. Histopathologic Examination

Sections from bone and cartilage were fixed in 10% formalin for 12 h, then decalcified in formic acid. The sections were processed in an automated tissue process (Leica, ASP 300S, Leica Biosystems, Richmond, IL, USA) and were paraffin embedded using a paraffin embedding system. Four-micrometer-thick sections were cut, and they were prepared and stained with hematoxylin and eosin stain (H&E) via an automated slide stainer system (Leica ST5010) for histopathological examination. Sections were examined under a light microscope.

### 2.5. Statistical Analysis

In this study, we analyzed raw data through GraphPad Prism 7.0 (San Diego, CA, USA). The statistical analysis was performed with one-way ANOVA, followed by Dunnett’s multiple comparison test. A *p*-value of under 0.05 was considered as significant and was indicated as * (*p* < 0.05), ** (*p* < 0.01), *** (*p* < 0.001), **** (*p* < 0.0001).

## 3. Results

### 3.1. Clinical Evaluation

#### 3.1.1. WOMAC Survey

The WOMAC score improved from a mean of 80.3 ± 6.7 preoperatively to 20.7 ± 4.8 at 6 months post-OP, and to 22.39 ± 7.7 at 12 M post-OP (Figure 2). The scores were significantly decreased from 3 M post-OP to 12 M post-OP.

#### 3.1.2. KOOS Survey

The score improved from a mean 50.15 ± 4.6 pre-OP to 80.38 ± 5.78 at 6 M post-OP, and it remained improved at 12 M post-OP with a mean of 79.16 ± 5.49 (Figure 3). The scores were significantly decreased from 3 M post-OP to 12 M post-OP.

### 3.2. Medical Imaging Analysis

#### 3.2.1. MRI Analysis and MOCART Score

MRI analysis showed the incorporation (T2) of the implanted grafts, with the surrounding tissues blurring the margins between them. In addition, the T1 image also showed the restoration of subchondral bone within the lesion (Figure 4). Five patients had a knee varus deformity of 5 to10 degrees at the standing mechanical axis, and five patients did not have a deformity. High tibial osteotomy was performed for the patients who had a varus deformity of more than 5 degrees with 3 degrees overcorrection. According to the X-ray images, all of the patients in Group B had their high tibial osteotomy healed by 12 weeks, and the MRI assessment was performed using the MOCART scoring criteria. The MOCART score showed a tendency to diminish from a mean of 69.3 ± 12.37 at 6 weeks to a mean of 82.85 ± 11.49 at 12 M post-OP (Figure 5).

#### 3.2.2. Arthroscopic Evaluation

The second look arthroscopy was performed during the removal procedure for the implanted plates for four patients, and one patient without high tibial osteotomy due to his complaint of feeling a loose body inside his knee. The arthroscopy showed a white hyaline-like tissue filling the whole articular cartilage defect and nearly homogenous with the native articular cartilage (Figure 6A,B).

### 3.3. Histopathologic Examination

A microscopic examination of sections stained using H&E stain revealed that intact cores formed of a cap of hyaline cartilage ranged in thickness over 2–2.6 mm (Table 2), except in one case where the cartilage was admixed with excess fibrous tissue and the thickness was 0.35 mm. In all biopsies, the cartilage was formed of chondrocytes dispersed within the extracellular matrix and residing in lacunae. Chondrocytes have pyknotic nuclei and lack abundant nuclear detail. Some chondrocytes were arranged in clusters (an early reaction to mechanical stress). Chondrocytes are flattened near the surface (superficial zone) and more rounded elsewhere, forming transitional, radial, and calcified zones. In addition, a thin basophilic line (tidemark) representing the boundary between unmineralized and mineralized zones was noted. The calcified zone is anchored to dense cancellous bone that entangles the bone marrow elements (Figure 6C,D).

## 4. Discussion

Aging is inevitable, and so is the decline in function of the various organ systems, particularly in synovial joints. Articular cartilage is prone to ‘wear and tear’, and the articular cartilage undergoes persistent degeneration. Osteoarthritis (OA) occurs when the rate of wear exceeds the rate of the regeneration of articular cartilage, resulting in chondral ulcers. OA is indeed one of the main causes of disability in older adults, affecting about 10% of men and 18% of women over the age of sixty. The universal increase in life expectancy makes OA one of the most important causes of disability [33]. Due to the poor capacity of regeneration of articular cartilage, most of the treatment methods used now are symptomatic treatments to decrease pain and disability caused by osteoarthritis. In this study, the trial of articular cartilage regeneration in knee osteoarthritis (KOA) was performed via a rather simple procedure proposed by ROKIT Healthcare, inc. The idea was to use MAT derived from adipose tissue mixed with allogenic hyaline cartilage powder derived from tissue banks and printed onto a 3D printed PCL mold. The mold was printed according to the measurements and shape of the knee articular cartilage defect. The graft was gelatinized by mixing with fibrin glue, then the defect was fixed using another package of fibrin glue. The fact that the adipose tissue was minimally manipulated through only micronization by passing through decreasing sizes of micronizers several times makes it a simple procedure. The whole process of bio-ink preparation, and the 3D printing of the mold and bio-ink took about 20–30 min inside the operative theater.

There have been few studies involving the use of MSCs in a 3D scaffold into human articular cartilage defects, such as that of Kuruda et al. in 2007 [34]. However, they mixed the bone marrow-derived stem cells into a collagen hydrogel, and they fixed it with a periosteal membrane. Still, they used it for a traumatic cartilage defect, and they also had to expand the cellular population in vitro for 4 weeks before the final procedure.

In this study, the 3D printed graft was composed of fat-derived MAT, with allogenic hyaline cartilage powder derived from treated costal cartilage in tissue banks and fixed using fibrin glue to the cartilage defect. MAT was derived from subcutaneous adipose tissue in a simple liposuction procedure, minimally manipulated on the operation table to be filtered and micronized. Then, it was printed onto a mold of PLC and printed using a Dr. INVIVO, a 3D printer, with the same size and shape of the patients’ articular cartilage defect. Since we used MAT derived from the patient’s autologous tissue, an immune rejection was not observed. In a previous study, a 3D printed patch consisting of MAT with lyophilized cartilage powder was transplanted to the osteochondral defects in beagle dogs. After 32 weeks of transplantation, the cartilage was regenerated with a smooth layer shape to form a cartilage-like matrix. In addition, the cartilage showed dense COL2 content within the de novo tissues, indicating its regeneration into hyaline-like cartilage [19]. In this study, an organized articular cartilage was found in the patient’s biopsy samples. The IMPACT trial assessed the safety and efficacy of a single-stage procedure for focal cartilage lesions in the knee using a combination of autologous chondrocytes and allogeneic BM-MSCs. In this study, 35 patients were treated using autologous ‘chondrons’ and allogeneic MSCs in a fibrin glue carrier. The 5-year-outcome data demonstrated that the majority of patients showed statistically significant and clinically relevant improvements in the KOOS and in all its subscales from baseline [35]. We believe that adding the PCL mold in our study helps with cartilage defect management due to proper fashioning to the exact defect size.

We divided the patients into two groups according to the addition of high tibial osteotomy for patients having varus malalignment to detect the effects of added procedures for osteoarthritis if needed, and to be able to harvest a biopsy during metal removal to avoid conflicts with the ethical considerations of performing a surgery for the sake of the research. In this study, patients’ pain and function improved, as shown from the improvement in three clinical scores. The WOMAC score improved from a mean of 80.3 ± 6.70 preoperatively to a mean of 22.39 ± 7.7 at 12 M post-OP. KOOS indices were improved from a mean 50.15 ± 4.60 preoperatively to a mean 79.16 ± 5.49 at one year postoperative. This improvement was observed in both groups.

The MRIs of patients were evaluated using the MOCART score, which is a specific score for articular cartilage graft assessment. The MOCART score improved from a mean of 69.3 ± 12.37 at 6 weeks (1.5 M) to a mean of 82.85 ± 11.49 after 12 M post-OP among all patients. In histopathology analysis, the grafted part of the articular cartilage showed a hyaline cartilage structure with characteristic chondrocytes (monocellular tissue) in abundant matrix when stained using H&E, showing the character of hyaline cartilage. The hyaline cartilage showed considerable thickness (more than 2 mm thickness), and a gradual transition of mineralization to the zone of cancellous bone. The fact that fibrous tissue was found in the only one case where the graft dropped inside the knee, and that the hyaline cartilage thickness was less than 0.4 mm adds more proof that the graft differentiated into hyaline cartilage. These results showed that the procedure can be beneficial in the treatment of moderate cases of knee osteoarthritis, whether alone or with other additive procedures such as high tibial osteotomy.

The benefits of a 3D printed graft containing MAT and allogenic hyaline cartilage matrix are that it is better than the monolayer chondrocyte implantation, which can have phenotype changes during cell proliferation, producing more fibrocartilage (types I, II, and III collagen) than hyaline cartilage (type II collagen). It is also completed in two stages, adding more risk to the patient, as well as more expenses. This procedure is a rather simple one-stage surgery. Hypothetically, it is more beneficial than the Osteochondral Autograft Transfer System (OATS), since the last is relatively contraindicated in osteoarthritis because the donor tissue is most likely affected by the degenerative process, affecting all of the articular cartilage of the knee [15].

The limitations of this procedure lie in the weak fixation at the articular cartilage, which is achieved using fibrin glue, which does not provide a solid fixation. As a matter of fact, one of our patients had a loss of the graft that led to the formation of loose bodies in the knee, which were removed arthroscopically after 11 months; the arthroscopic picture of the articular cartilage defect showed partial healing, and the histologic examination showed fibrous tissue formation.

## 5. Conclusions

The novel procedure using the 3D printing of a mixture of MAT with allogenic hyaline cartilage matrix on a mold of the PCL that was printed with the same size and shape of the articular cartilage defect can be beneficial in the treatment of moderate cases of knee osteoarthritis. It is a rather simple one-stage procedure with cost efficiency.

## Figures and Tables

**Figure 1 jpm-13-00748-f001:**
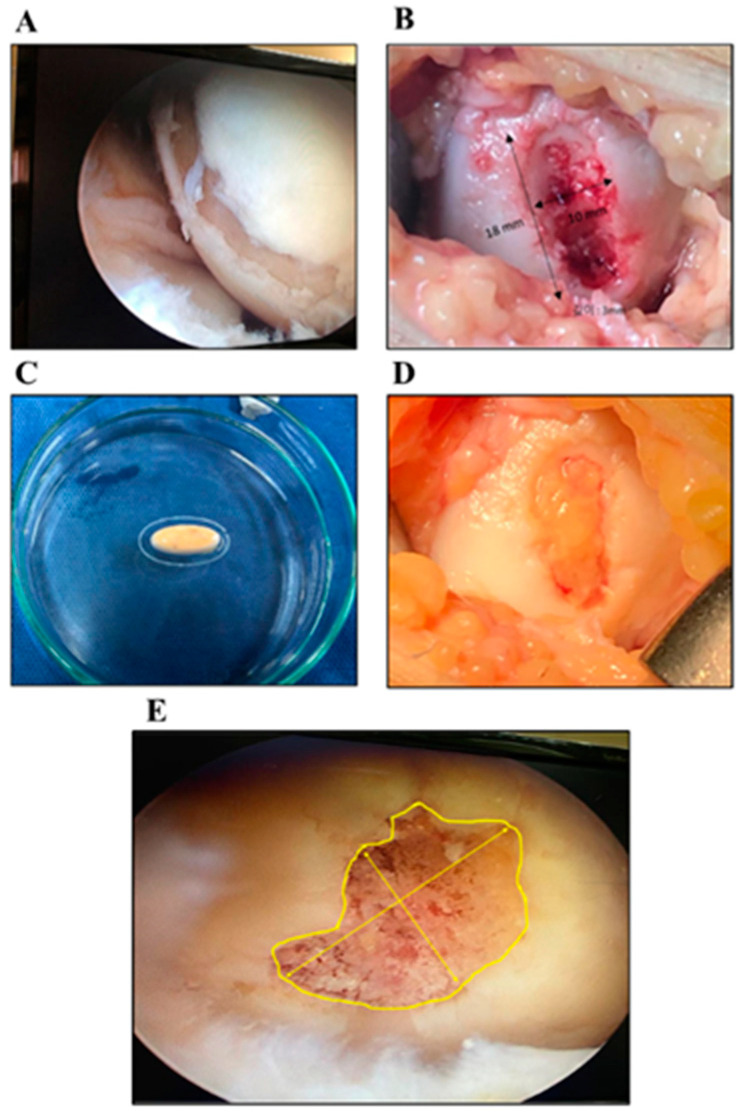
Representative Images for the Surgery Procedures. An arthroscopic view of the osteochondral lesion (**A**); Gross appearance of the ulcer at medial condyle (**B**); A 3D bioprinted LCCM/MAT graft (**C**); The fixed graft within the osteochondral lesion (**D**); A representative image of measuring osteochondral lesions to generate STL file (**E**).

**Figure 2 jpm-13-00748-f002:**
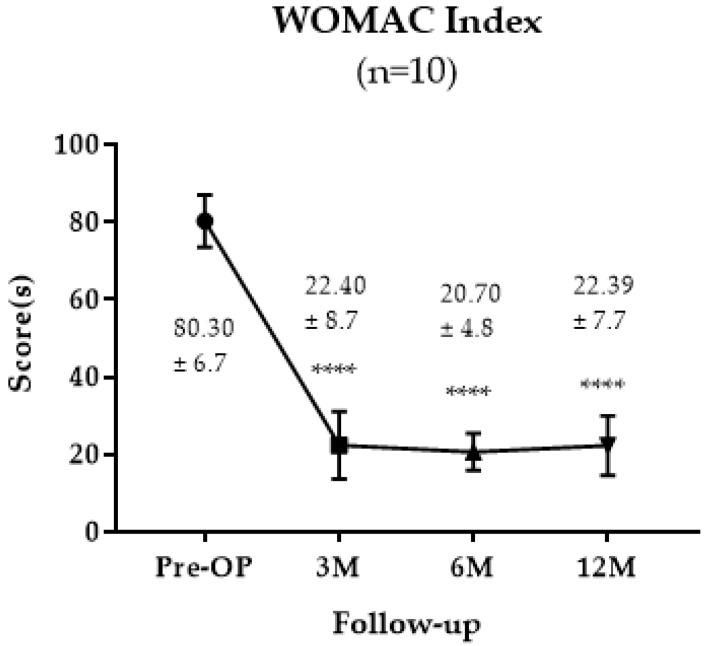
WOMAC Survey Results of the Participants. The significance was indicated as **** (*p* < 0.0001). The data are described as mean ± SD.

**Figure 3 jpm-13-00748-f003:**
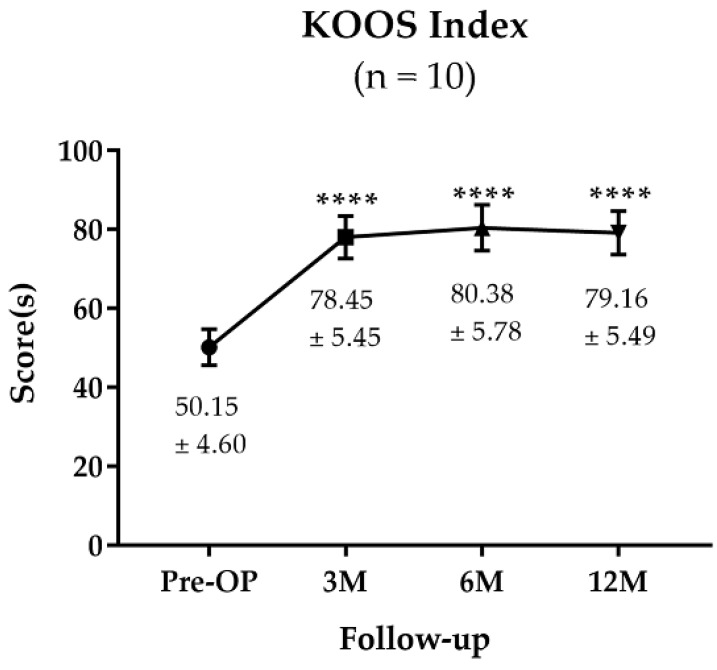
KOOS Survey Results of the Participants. The significance was indicated as **** (*p* < 0.0001). The data are described as mean ± SD.

**Figure 4 jpm-13-00748-f004:**
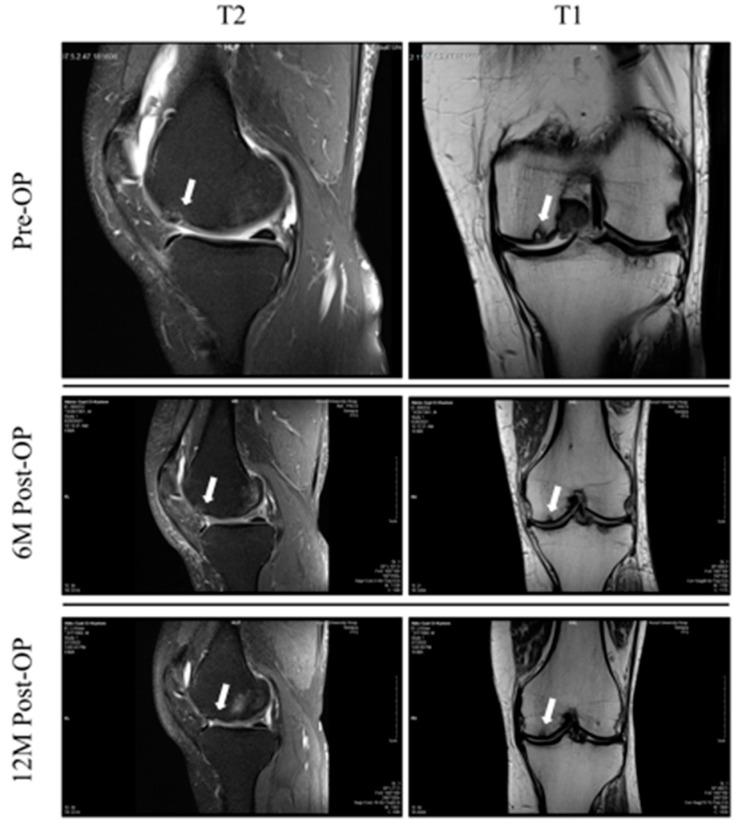
Magnetic resonance images of the participants from pre-OP to 12 M post-OP follow-up. T2 images are placed in left column and T1 images are placed in right column. Defect site and gradual graft healing over time are indicated by white arrows.

**Figure 5 jpm-13-00748-f005:**
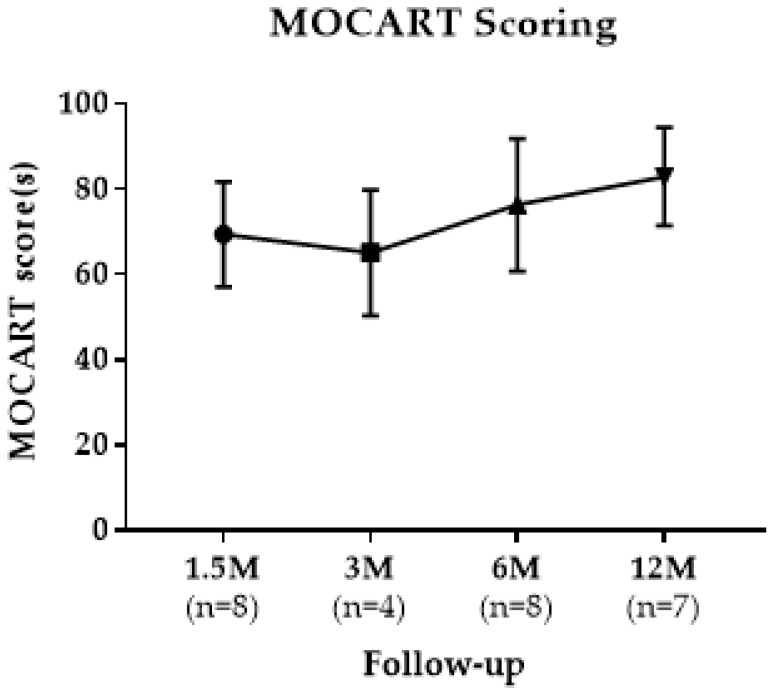
MOCART Scoring Results of the Participants from 3 M post-OP to 12 M post-OP. The data are described as mean ± SD.

**Figure 6 jpm-13-00748-f006:**
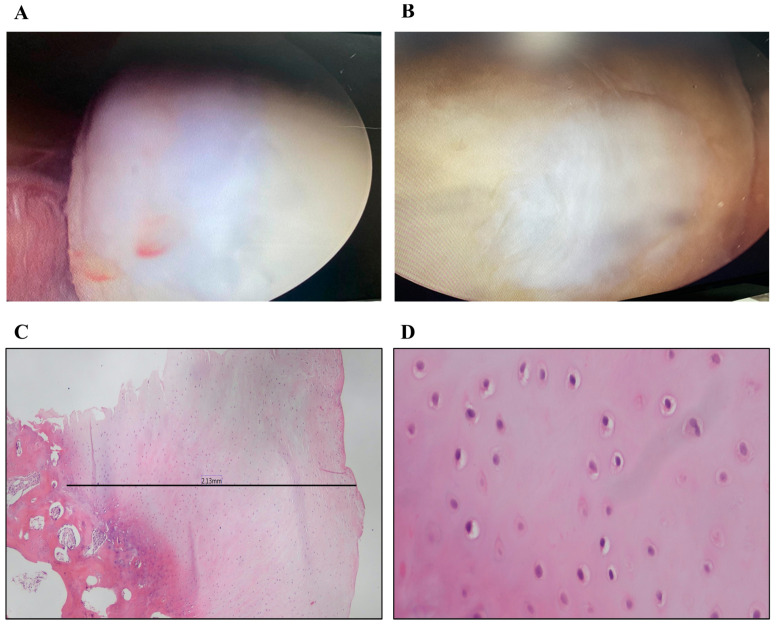
Arthroscopic Images of Knee Cartilages and Histopathology Examination at 12 M post-OP Follow-up. A second look arthroscopic image showing white de novo cartilage-like tissue (**A**); A closer image showing lesion site filled with hyaline-like cartilage (**B**); A representative histopathologic image showing de novo hyaline-like cartilage, with the cartilage thickness at defect site as indicated by black line (**C**); A magnified (40×) image showing differentiated chondrocytes within lacuna (**D**).

**Table 1 jpm-13-00748-t001:** Patient characteristics.

Age	Number of Patients
>18–20 years old	1
40–50 years old	8
>50 years old	1
Gender	Number of Patients
Male	5
Female	5
Body Mass Index	Number of Patients
>18.5–24.9 kg/m^2^	4
25–29.9 kg/m^2^	5
30–34.9 kg/m^2^	1

**Table 2 jpm-13-00748-t002:** Histopathology analysis results.

Patient Number	De novo Cartilage (mm)	Fibrous Tissue (mm)
3	2.20–2.60	- ^1^
4	2.12	0.4
7	2.13–2.30	0.3
8	2.0–2.60	-
10	0.35	0.65

^1^ Not observed.

## Data Availability

Data is unavailable for ethical restrictions.

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
