# Peer review of "The Evaluation of Cartilage Regeneration Efficacy of Three-Dimensionally Biofabricated Human-Derived Biomaterials on Knee Osteoarthritis: A Single-Arm, Open Label Study in Egypt"

_jpm, 2023, doi:10.3390/jpm13050748_

Round 1

Reviewer 1 Report

GENERAL

I want to congratulate the authors for their efforts put into writing a cohesive, clear, and well-structured manuscript.

The manuscript needs three mandatory changes:

1.       Revise grammatical mistakes to improve the overall quality of the manuscript.

2.       Revise figure citations, abbreviations, and some definitions.

3.       Improve “Histopathologic examination” data.

SPECIFIC

The subject is missing in the third sentence of the manuscript (abstract):

In this study, to assess the short-term clinical outcome of a novel surgical technique that uses 3D bioprinted micronized adipose tissue (MAT) graft for knee cartilage defects as well as evaluation of the degree of incorporation of such graft types at the arthroscopic and radiological analysis.

Could be (or in past tense):

In this study, we assess….

Another one:

Its pathogenesis includes detrimental alterations on the articular cartilage of the joint including synovial inflammation, cartilage matrix degradation, and following exposure of subchondral bone tissues.

Could be:

Its pathogenesis includes detrimental alterations on the articular tissues of the joint

These grammatical mistakes are found throughout the manuscript. Please revise the manuscript.

The use of WOMAC, KOOS, and MOCART is correct for implant integration. No score can evaluate the MAT-hyaline cartilage matrix -polycaprolactone stabilization, patient biocompatibility, and short-term & long-term integration. Please mention alternative evaluations.

This is a medical science manuscript, please, use the word “regeneration” carefully. It is different from repair, integration, resolution…etc… Authors may define “regeneration” in the text.

Authors usually need to define novel abbreviations to define novel terms. Nonetheless, universally approved abbreviations should never be replaced. As far as the reviewer knows, TC2 is not a suitable abbreviation for type II collagen. COLII, Colll, COL2, or Col2 are the standard abbreviations each one of them defining human protein, mice gene…mutants….etc…

To have consistency across the manuscript, please define MAT in the introduction too.

Polycaprolactone: please add some information regarding this critical component in the technique. More specifically, why is it better than alternatives?

“Body mass index of less than 35 were included”. Did you include patients with overweight?

Please add the following columns in Table 1: Gender and patient group.

There are many handicaps to these techniques but overall problems usually arise from poor attachment (fibrin glue) and tissue integration (PLC-MAT). Could you provide a brief explanation of the use of fibrin glue and its characteristics?

Moreover, Figure 1C shows the 3D-bioprinted LCCM/MAT graft with smooth and blunt edges which turn on partially representing the ulcer. Is it a limitation of the techniques (resolution)? Is It on purpose? Please incorporate a short sentence regarding the 3D reconstruction by software.

Please be more precise and extend in this particular sentence “The wound is closed in layers, and the dressing was applied”

Are there any osteogenic, chondrogenic, or anti-adipogenic factors included in the graft? The point here is to assess what are the authors expecting to happen with the MSC.

Please calls to pictures should maintain style cohesion. Each individual Figure of a Figure panel is mandatory to be individually labeled. Therefore, Figure 4 should be labeled as Figure 4A, Figure 4B…and so on

Similarly, please use MRI instead of MR.

Please, reduce the grade of certainty in this sentence: “MR analysis showed proper incorporation (T2) of the implanted grafts with surrounding tissues blurring the margins between them.

If Figure 6 call comes before Table 2, that is also the order of displaying in the manuscript.

Overall the manuscript is interesting, detailed, and well-documented… until we reach Figures 6C and 6D. More solid data is needed to demonstrate complete or partial incorporation of the MAT. Furthermore, there are many stainings better suited than H&E to evaluate cartilage fibers, bone, or fat. Please provide more and better images. Remember, that ALL statements need to be supported by data. Please, indicate the position of MAT in these pictures using an arrow or similar. I would encourage the authors to do the same across all pictures. These changes are mandatory.

I want to congratulate the authors again for advancing science and caring for patients.

Author Response

All comments are appreciated and be considered to improve the manuscript clarity and quality

“Moreover, Figure 1C shows the 3D-bioprinted LCCM/MAT graft with smooth and blunt edges which turn on partially representing the ulcer. Is it a limitation of the techniques (resolution)? Is It on purpose? Please incorporate a short sentence regarding the 3D reconstruction by software”.

 we debrided the edge to more regular shape that transformed by the software to a smooth printed graft. We used this graft to fit the defect precisely.

“The use of WOMAC, KOOS, and MOCART is correct for implant integration. No score can evaluate the MAT-hyaline cartilage matrix -polycaprolactone stabilization, patient biocompatibility, and short-term & long-term integration. Please mention alternative evaluations.”

This study focused on clinical, radiological and histopathological assessment and did not concern with MAT-hyaline cartilage matrix -polycaprolactone stabilization, patient biocompatibility, and short-term & long-term integration.

“Are there any osteogenic, chondrogenic, or anti-adipogenic factors included in the graft? The point here is to assess what are the authors expecting to happen with the MSC. “

No

All other comments are corrected within the text

Reviewer 2 Report

Dear Authors, it was a pleasure to see the results of your clinical study. The idea and the proposed solution are very interesting and the results seem promising. However, I have some comments, on how your manuscript could be improved:

1.       In this study, 10 patients were divided into two groups based on the alignment of the knee. Furthermore, they received slightly different cartilage management techniques, thus do authors agree, that all the results, should be displayed for each group separately? Currently, in the graphs and tables, there are no separation of the groups. Thus, all the graphs should be remade, to indicate each group's results.

2.       In the method section 2.2. there is no information on how PCL and MAT constructs were designed. Please provide the full process, the software used, the printing nozzle diameter, the printing temperature, etc. Furthermore, there is no information on where allogenic hyaline cartilage powder, fibrin glue, and PCL was bought.

3.       Authors should include at least one image of STL (or other 3D format files) of a patient PCL mold and MAT to better illustrate how it was created. Also, Authors should explain in more detail if the construct: PCL mold with MAT, or only MAT is implanted in the cartridge defect. These figures should be appended to fig. 1.

4.       In the statistical analysis section, Authors state that they used ANOVA followed by Dunnett’s multiple comparison tests to analyse the data. This is not the best choice for this situation, since it only compares differences to control (this case time point 0) to other time points, but it will not test differences between the time points. Also, if authors will agree to separate groups' results, they will require to compare differences in different time points between the groups. Here is a good article which explains which multiple comparisons post hoc tests are appropriate in different scenarios: https://www.ncbi.nlm.nih.gov/pmc/articles/PMC6193594/

5.       Figure 4 has no explanation of the images. In the caption, Authors should explain what is T1 and T2. Also, if patients will be divided into two groups, each group should have its representative images. Furthermore, as it is a human study, if possible, authors should include all the MRI images. It can be added to the supplement file. Also, please use arrows to show where are defects and scaffolds located.

6.       Figure 6. I have the same request, to include more images of the patients in the graphs. Also, please upload all the histological images of each patient in the supplement material file. Furthermore, please use arrows to show where are defects and scaffolds located.

7.       In section 3.2.1. Authors state that the MOCART score improved from a mean of 69.3 ±12.37 at 6 weeks to a mean of 82.85 ± 11.49 at 12M post-OP (Figure 4.). However, in the figure, there are no significant differences between time point 0 and other time points. Thus, Authors cannot state that the MOCART score improved, they can only say that they observed a tendency in diminishing MOCART score.

8.       The discussion is too long and has a lot of information which could be used in the introduction section, but not in the discussion, e.g.:

Although the last therapeutic option has been shown to relieve pain and improve
mobility in people, it may carry several complications like stiffness, instability, aseptic
loosening, infection, prosthesis failure, and mal-alignment [26]. In the last few decades
great ambitions were laid upon regenerative medicine as an alternative method for tradi-
tional surgical procedures. The idea is that the regenerated cartilage tissue should behave
chemically and mechanically as the native articular cartilage [27]. The most widely used
method for articular cartilage regeneration is the autologous chondrocyte implantation
(ACI). The procedure was first introduced by the Swedish group in 1994. The procedure
consisted in the removal of small biopsies of cartilage (200/300 mg) from less loaded areas
of the knee. Isolated chondrocytes were cultured in monolayer and after 2 – 3 weeks, har-
vested as a suspension for implantation. The suspension was then injected into the defect
under a periosteal flap derived from the proximal medial tibia (first-generation ACI) [28].
Last generation ACI, also referred to as matrix induced ACI (MACI), comprises the seed-
ing of in vitro expanded chondrocytes onto a three-dimensional (3D) scaffold. The engi-
neered tissue is subsequently trimmed to the size of the defect and implanted with fibrin
glue fixation [29]. To our knowledge it was used in humans for knee OA only once in 2010
by Means et., al [30], and they found that it decreased the pain, and improved the function
as in traumatic cases. However, the ACI was considered contraindicated in OA, since the
degenerative microenvironment may cause the implanted chondrocytes to undergo un-
desired dedifferentiation or apoptosis, therefore undermining efficacy. Also, the ACI procedure is considered long, complicated, and expensive. It also needs two hospitaliza-
tion events for the patient one for the biopsy harvest, and one for the implantation.
In the last few decades attempts of regeneration of articular cartilage was done by
the use of Mesenchymal Stem Cells (MSCs) injected intra-articular. They are easy to iso-
late, and do not imply ethical problems [31]. Lately, stem cells derived from subcutaneous
adipose tissue (ASCs) have been considered less invasive procedure, and suitable for ap-
plication in regenerative therapies. The advantage over Marrow SCs is that ASCs are pre-
sent in a higher percentage (500 folds more) [32]. The combination of stem cells and the
chondron (hyaline cartilage matrix) was used in a goat model in 2013. They found that
those treated with the chondron/MSC treatment were almost filled completely with carti-
lage-like tissue in comparison to those treated by microfracture where only 25 - 50% of the
defect was filled by a fibrocartilage tissue after 6 months [33].

9.       In the discussion Authors again state that they divided patients into two groups, however, they have not provided any information about the differences between the groups observed.

10.   This paragraph is missing citations:

The benefits of 3D printed graft containing MAT and allogenic hyaline cartilage ma-
trix are that it is better than the monolayer chondrocyte implantation which can have phe-
notype changes during cell proliferation producing more fibrocartilage (type I, II, and III
collagen) than hyaline cartilage (type II collagen). It is also done in two stages adding more
risk on the patient and more expenses. This procedure is rather simple one stage surgery.
Hypothetically, it is more beneficial than Osteochondral Autograft Transfer System
(OATS), since the last is relatively contraindicated in osteoarthritis because the doner tis-
sue is most likely affected by the degenerative process affecting all the articular cartilage
of the knee.

11.   Authors should compare their results to other researchers' published results. Is their method superior? What are the results if isolated MSC is used. Also, how their PCL mold helps to improve the surgery flow. Authors can find come clinical data in these reviews:

https://stemcellres.biomedcentral.com/articles/10.1186/s13287-021-02689-9

https://pubmed.ncbi.nlm.nih.gov/34423830/

Author Response

  1. As regards the surgical treatment choice for the patients included we were not aiming to compare the results of treating cartilage defects with and without high tibial osteotomy , our aim was to exclude the effect of deformity as a contributing factor in outcome in order to include more patients in our series so we converted patients with deformed knees into normal alignment via osteotomy and kept the cartilage management strategy the same . Results of osteotomy has been thoroughly discussed in literature and we used it as a tool of obtaining normal alignment and that is why we displayed the results as one group that included ultimately non deformed knees without osteotomy in normal knees and after osteotomy in varus knees .
  2. A . PCL
  • PCL (Polycaprolactone) filament was purchased from Evonik (Essen, Germany).
  • For PCL fabrication, we used NewCreaterK software (ROKIT Healthcare, Seoul, Republic of Korea).
  • For PCL fabrication, the nozzle size is 0.8mm, and temperature is set to be 180 Celsius
  1. Bioink
  • Allogenic hyaline cartilage powder : In case of AHC, the donated human costal cartilages were purchased from Korean tissue bank and  lyophilized for 24 hours and then pulverized using liquid nitrogen and Retsch ball miller (Retsch, Hann, Germany) into 200um diameter in average.
  • In case of fibrin glue, we purchased Beriplast P Combi Set (CSL Behring, King of Prussia, PA, USA) for scaffold solidification and attachment.
  • For biofabrication of bioink, we used 21-gauze blunt needle.
  1. Added in mauscript
  2. As forementioned in point 2 we chose to display the patients as one group after correction of deformity by osteotomy and this is the explanation for using our statistical analysis methods.
  3. Added in manuscript + supplementary files added
  4. Added in manu script + supplementary files added
  5. Corrected in manuscript
  6. Corrected in manuscript
  7. We divided the patients into two groups as a description of the surgical technique without the aim of comparison of outcome between group A and B.
  8. Added in manuscript
  9. Added in manuscript
